# Clinically Distinct Subtypes of Acute Kidney Injury on Hospital Admission Identified by Machine Learning Consensus Clustering

**DOI:** 10.3390/medsci9040060

**Published:** 2021-09-24

**Authors:** Charat Thongprayoon, Pradeep Vaitla, Voravech Nissaisorakarn, Michael A. Mao, Jose L. Zabala Genovez, Andrea G. Kattah, Pattharawin Pattharanitima, Saraschandra Vallabhajosyula, Mira T. Keddis, Fawad Qureshi, John J. Dillon, Vesna D. Garovic, Kianoush B. Kashani, Wisit Cheungpasitporn

**Affiliations:** 1Division of Nephrology and Hypertension, Department of Medicine, Mayo Clinic, Rochester, MN 55905, USA; ZabalaGenovez.Jose@mayo.edu (J.L.Z.G.); kattah.andrea@mayo.edu (A.G.K.); Qureshi.Fawad@mayo.edu (F.Q.); dillon.John@mayo.edu (J.J.D.); garovic.Vesna@mayo.edu (V.D.G.); kashani.Kianoush@mayo.edu (K.B.K.); 2Division of Nephrology, University of Mississippi Medical Center, Jackson, MS 39216, USA; pvaitla@umc.edu; 3Department of Internal Medicine, MetroWest Medical Center, Tufts University School of Medicine, Boston, MA 01702, USA; voravech.niss@gmail.com; 4Division of Nephrology and Hypertension, Department of Medicine, Mayo Clinic, Jacksonville, FL 32224, USA; mao.michael@mayo.edu; 5Department of Internal Medicine, Faculty of Medicine, Thammasat University, Pathum Thani 12121, Thailand; pattharawin@hotmail.com; 6Section of Cardiovascular Medicine, Department of Medicine, Wake Forest University School of Medicine, Winston-Salem, NC 27101, USA; svallabh@wakehealth.edu; 7Division of Nephrology and Hypertension, Department of Medicine, Mayo Clinic, Phoenix, AZ 85054, USA; keddis.Mira@mayo.edu

**Keywords:** acute kidney injury, AKI, clustering, machine learning, nephrology, artificial intelligence, mortality, hospitalization

## Abstract

Background: We aimed to cluster patients with acute kidney injury at hospital admission into clinically distinct subtypes using an unsupervised machine learning approach and assess the mortality risk among the distinct clusters. Methods: We performed consensus clustering analysis based on demographic information, principal diagnoses, comorbidities, and laboratory data among 4289 hospitalized adult patients with acute kidney injury at admission. The standardized difference of each variable was calculated to identify each cluster’s key features. We assessed the association of each acute kidney injury cluster with hospital and one-year mortality. Results: Consensus clustering analysis identified four distinct clusters. There were 1201 (28%) patients in cluster 1, 1396 (33%) patients in cluster 2, 1191 (28%) patients in cluster 3, and 501 (12%) patients in cluster 4. Cluster 1 patients were the youngest and had the least comorbidities. Cluster 2 and cluster 3 patients were older and had lower baseline kidney function. Cluster 2 patients had lower serum bicarbonate, strong ion difference, and hemoglobin, but higher serum chloride, whereas cluster 3 patients had lower serum chloride but higher serum bicarbonate and strong ion difference. Cluster 4 patients were younger and more likely to be admitted for genitourinary disease and infectious disease but less likely to be admitted for cardiovascular disease. Cluster 4 patients also had more severe acute kidney injury, lower serum sodium, serum chloride, and serum bicarbonate, but higher serum potassium and anion gap. Cluster 2, 3, and 4 patients had significantly higher hospital and one-year mortality than cluster 1 patients (*p* < 0.001). Conclusion: Our study demonstrated using machine learning consensus clustering analysis to characterize a heterogeneous cohort of patients with acute kidney injury on hospital admission into four clinically distinct clusters with different associated mortality risks.

## 1. Introduction

Acute kidney injury (AKI) is a common medical complication of acute illnesses that affects approximately 10–20% of all hospitalized patients, with an even higher incidence in intensive care units [1,2,3]. AKI is associated with significantly increased mortality, length of hospital stay, and health care expenditure [4]. AKI can be broadly classified based on its severity, mechanisms (e.g., nephrotoxicity, hemodynamic, inflammatory), histology (e.g., glomerular, interstitial, tubular), or its etiology (e.g., sepsis-associated acute kidney injury (S-AKI), cardiac surgery-associated acute kidney injury (CSA-AKI), urinary obstruction, or contrast-associated AKI) [5,6,7]. Identifying the phenotype of patients with AKI on hospital admission can often be challenging due to limited clinical details before admission.

With the advancement of the electronic health record (EHR), machine learning (ML) approaches have been utilized to assist in clinical decision-making processes [8,9,10,11,12,13]. Consensus clustering is an unsupervised ML technique used to identify novel data patterns [14]. It can search for similarities and heterogeneities among large categories of data variables and isolate them into clinically meaningful clusters [8,15,16,17]. Recent studies have shown that disease subtypes determined by ML clustering methods can forecast different clinical outcomes [18,19].

This study attempts to identify clinically relevant clusters of patients with community-acquired AKI (CA-AKI) and assess their association with in-hospital and one-year mortality risks.

## 2. Materials and Methods

### 2.1. Patient Population

The Mayo Clinic Institutional Review Board approved this study (IRB number: 21-004248). We identified all adults (≥18 years) admitted to Mayo Clinic Hospital in Rochester, Minnesota, from January 2011 to December 2013. We only analyzed the first admission if patients had more than one hospital admission during the study period. We included patients who had AKI at hospital admission, i.e., CA-AKI. We identified and staged AKI according to the serum creatinine criteria of the 2012 Kidney Disease: Improving Global Outcome (KDIGO) guidelines. To identify CA-AKI, we used the most recent outpatient serum creatinine before hospital admission as the baseline level and compared it with the first serum creatinine measured within 24 h of hospital admission. CA-AKI was adjudicated when the admission serum creatinine was ≥0.3 mg/dL or 1.5-time higher than the baseline value. We excluded 1) patients who did not have serum creatinine measurement within 24 h of hospital admission or outpatient serum creatinine before hospital admission, 2) end-stage kidney disease patients on chronic dialysis or chronic kidney disease stage 5 patients with baseline estimated glomerular filtration rate (eGFR) of <15 mL/min/1.73 m^2^, and 3) patients who did not provide the Minnesota research authorization.

### 2.2. Data Collection

We collected pertinent information, including clinical characteristics and laboratory data, from our hospital’s electronic database using previously validated methods [20,21,22,23,24]. We aimed to categorize patients with CA-AKI into clusters based on their admission demographic information, principal diagnoses, comorbidities, and laboratory data. We only utilized available data within 24-h of hospital admission for clustering analysis. We selected the first laboratory value within this 24-h time frame when there were multiple measurements. We excluded variables with >20% missing data. If missing data were <20%, we imputed missing data using Random Forest multiple imputations before their inclusion into clustering analysis. Random Forest applies bootstrap aggregation of multiple regression trees to reduce the risk of overfitting and combines estimates from many trees [25].

The outcomes were hospital mortality and 1-year mortality. We initiated follow-up from hospital admission until death or 1-year after hospital admission and censored their follow-up at the date of their last inpatient/outpatient follow-up visit. We determined patient death and their death date using our hospital’s registry and Social Security Index.

### 2.3. Clustering Analysis

We conducted unsupervised consensus clustering analysis to develop clinical clusters of CA-AKI patients [26]. We used a pre-specified subsampling parameter of 80% with 100 iterations and assigned the number of potential clusters (k) to range from 2 to 10 to avoid excessive numbers of clusters that would not be clinically useful. The optimal number of clusters was determined by examining the consensus matrix (CM) heat map, cumulative distribution function (CDF), cluster-consensus plots in the within-cluster consensus scores, and the proportion of ambiguously clustered pairs (PAC) [27,28]. The within-cluster consensus score, ranging between 0 and 1, is defined as the average consensus value for all pairs of individuals belonging to the same cluster [28]. A value closer to one indicates better cluster stability [28]. PAC, ranging between 0 and 1, is calculated as the proportion of all sample pairs with consensus values falling within the predetermined boundaries [27]. A value closer to zero indicates better cluster stability [27]. We calculated the PAC using two criteria 1) the strict criteria consisting of a predetermined boundary of (0, 1), where a pair of individuals who had a consensus value >0 or <1 was considered ambiguously clustered, and 2) the relaxed criteria consisting of a predetermined boundary of (0.1, 0.9), where a pair of individuals who had consensus value >0.1 or <0.9 was considered ambiguously clustered [27]. The detailed consensus cluster algorithms are provided in the online Appendix A.

### 2.4. Statistical Analysis

After we categorized eligible patients into clusters using the described unsupervised ML approach, we assessed differences in clinical characteristics and outcomes among the clusters. We tested for differences in baseline characteristics between the identified clusters using the analysis of variance (ANOVA) test for continuous variables and the Chi-squared test for categorical variables. We calculated the standardized mean differences of individual baseline characteristics between each cluster and the overall population for the clusters’ profile analysis. We considered variables with an absolute standardized mean difference of >0.3 as key features of the cluster. In addition, we compared hospital and one-year mortality as outcomes of interest across the identified clusters. We assessed hospital mortality risk using logistic regression and one-year mortality using Cox-proportional hazard regression analysis. We utilized cluster 1 as the reference group for mortality comparison, as this cluster was associated with the lowest mortality. We did not adjust for between-group differences in baseline characteristics because all of these variables were utilized to develop the identified clusters through unsupervised ML. We performed all analyses using R, version 4.0.3 (RStudio, Inc., Boston, MA, USA; Available online: http://www.rstudio.com/ (accessed on 15 July 2021)), with the packages of ConsensusClusterPlus (version 1.46.0) [28] for consensus clustering analysis and the missForest package for missing data imputation [25].

## 3. Results

A consensus clustering analysis was performed in a total of 4289 hospitalized patients with CA-AKI. The mean age was 67 ± 16 years, and 60% were male. For AKI severity, 82% had stage 1, 10% had stage 2, and 8% had stage 3 AKI.

The CDF plot displays consensus distributions for each AKI cluster (Figure 1A). The delta area plot shows the relative change in the area under the CDF curve (Figure 1B). The most considerable changes in the area occurred between k = 3 and k = 5, at which point the relative increase in areas became noticeably smaller. As shown in the CM heatmap (Figure 1C, Appendix A), the ML algorithm identified four clusters with clear boundaries (Figure 1C), indicating good cluster stability over repeated iterations.

There were 1201 (28%) patients in cluster #1, 1396 (33%) patients in cluster #2, 1191 (28%) patients in cluster #3, and 501 (12%) patients in cluster #4. Table 1 shows the baseline characteristics of the four identified clusters.

Cluster #4 had the highest mean cluster consensus score, representing high stability (Figure 2A). Favorable low PACs by both strict and relaxed criteria were demonstrated in all clusters (Figure 2B). Each identified cluster encompassed patients with distinct baseline characteristics.

Figure 3 shows the plot of standardized mean difference to visualize the key features of each cluster. Cluster #1 included younger patients with lower serum potassium and comorbidity burden, particularly congestive heart failure and diabetes mellitus, and higher eGFR and hemoglobin. Cluster #2 patients were older and had lower eGFR, serum bicarbonate, strong ion difference, and hemoglobin, but higher serum chloride. Similar to cluster #2, cluster #3 patients were also older and had a lower eGFR. In contrast, cluster #3 patients had lower serum chloride but higher serum bicarbonate and strong ion difference. Congestive heart failure was more common in cluster #3 patients as well. Cluster #4 patients were younger and more likely to be admitted for genitourinary and infectious diseases but less likely for cardiovascular diseases. The most prominent key feature of cluster #4 patients was the higher severity of CA-AKI. Most cluster #4 patients had AKI stage 3 (66%), followed by stage 2 (25%) and stage 1 (9%), while patients in the other clusters more frequently had AKI stage 1 at hospital presentation. Other key laboratory features of cluster #4 patients included lower serum sodium, chloride, and bicarbonate but higher serum potassium and anion gap.

Hospital mortality was 1.7% for cluster #1, 4.4% for cluster #2, 3.9% for cluster #3, and 11.2% for cluster #4 (*p* < 0.001) (Figure 4A). Clusters #2, #3, and #4 had higher hospital mortality compared to cluster #1, with odds ratios of 2.74 (95% CI 1.65–4.57), 2.37 (95% CI 1.39–4.04), and 7.43 (95% CI 4.41–12.53), respectively (Table 2). The median follow-up time was 1.1 (IQR 0.3–2.1) years. One-year mortality was 8.4% for cluster #1, 29.7% for cluster #2, and 31.2% for cluster #3, and 33.7% for cluster #4 (*p* < 0.001) (Figure 4B).

Clusters #2, #3, and #4 had higher one-year mortality compared to cluster #1, with hazard ratios of 3.97 (95% CI 3.14–5.03), 4.22 (95% CI 3.33–5.35), and 4.98 (95% CI 3.82–6.48), respectively (Table 2).

## 4. Discussion

In this study, the unsupervised consensus clustering approach was utilized to categorize patients with CA-AKI into unique clusters. This produced four clinically meaningful clusters with high cluster stability. The four clusters demonstrated dissimilar patients’ characteristics and were associated with different hospital and one-year mortality risks.

Cluster #4 had the highest in-hospital and one-year mortality. This cluster consists of patients with the lowest sodium level, highest potassium level, and severe acid-base disturbances as manifested by the lowest bicarbonate level associated with the highest anion gap, compared to other clusters. Patients in this cluster also suffered from more severe AKI than other clusters. AKI and the associated electrolyte imbalances and acid-base disorders may have contributed to the observed increased mortality risk [29,30,31,32]. In addition, they were more likely to have comorbidities, including diabetes mellitus and cirrhosis, which have been shown in prior studies to portend a worse prognosis.

Cluster #1 had the lowest in-hospital and one-year mortality compared to other clusters. Patients in cluster #1 had the lowest mean age and comorbidity burden while simultaneously having the highest eGFR and hemoglobin, which could explain the lowest mortality in this cluster. Patients in clusters #2 and #3 had in-hospital and one-year mortality rates between clusters 1 and 4. Compared to the average of all AKI patients, AKI in cluster #2 occurred in the settings of lower serum bicarbonate, strong ion difference, and hemoglobin, but higher serum chloride. AKI in cluster #3 occurred with lower serum chloride but higher serum bicarbonate and strong ion difference. Patients in cluster #3 more commonly presented with a principal diagnosis of cardiovascular diseases on admission. It could be hypothesized that cluster #2 represented AKI patients with non-anion gap metabolic acidosis with hemoconcentration, while patients in cluster #3 had contraction alkalosis due to diuresis from heart failure. The majority of AKI in clusters #1, #2, and #3 were stage 1 AKI. Although clusters #2 and #3 were older and had higher comorbidities than those in cluster #4, patients in clusters #4 carried higher in-hospital and one-year mortality risks. This is likely due to the significant impacts of AKI severity, infectious disease, and hyponatremia on poor outcomes [32,33]. The findings of our study suggest that the use of the ML approach may help identify potential new pathophysiological pathways leading to CA-AKI.

Our study has several limitations. First, this is a single-center study, and 94% of the study population is White. In addition, as this study was conducted using the data of hospitalized patients from 2011 to 2013, future studies using a more up-to-date dataset is suggested to confirm our finding. Second, the ML clustering approach was performed at hospital admission to allow application of this research for future clinical practice, where early recognition of patient mortality risk would allow for earlier intervention via prevention and treatment. However, our identified clusters included CA-AKI patients without considering the exposures during hospitalization, including hospital-acquired infections and procedural or medication-related adverse events. Nevertheless, our ML clustering approach successfully identified four clusters among CA-AKI patients with distinct novel phenotypes that indicated different in-hospital and one-year mortality rates. Future studies are required in a more diverse population and health systems to evaluate the discriminatory ability of the clusters we identified. Studies to investigate the underlying mechanisms for the identified modifiable phenotypical features to potentially improve outcomes through precision medicine are necessary.

## 5. Conclusions

In conclusion, this is the first study utilizing ML consensus clustering analysis of hospitalized patients with admission AKI. Our findings suggest four distinct phenotypic and clinicopathological clusters of admission AKI with different in-hospital and one-year mortality risks. The highest mortality risk of AKI on admission was observed among patients with higher AKI severity, hyponatremia, metabolic acidosis, and a principal diagnosis of infectious disease. These findings may potentially help classify hospitalized patients with AKI on admission which are associated with different mortality risks, and translate towards an improved personalized medicine approach.

## Figures and Tables

**Figure 1 medsci-09-00060-f001:**
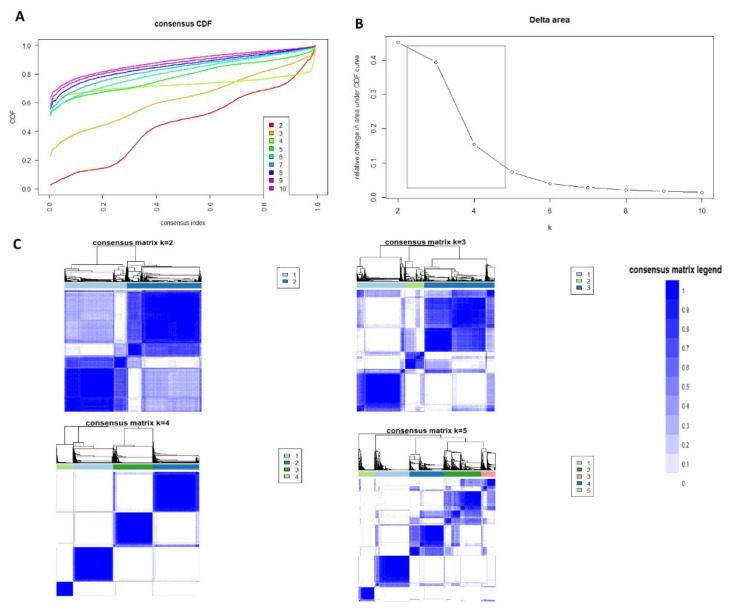
(**A**) CDF plot displaying consensus distributions for each k; (**B**) Delta area plot reflecting the relative changes in the area under the CDF curve. (**C**) Consensus matrix heat map depicting consensus values on a white to blue color scale of each cluster.

**Figure 2 medsci-09-00060-f002:**
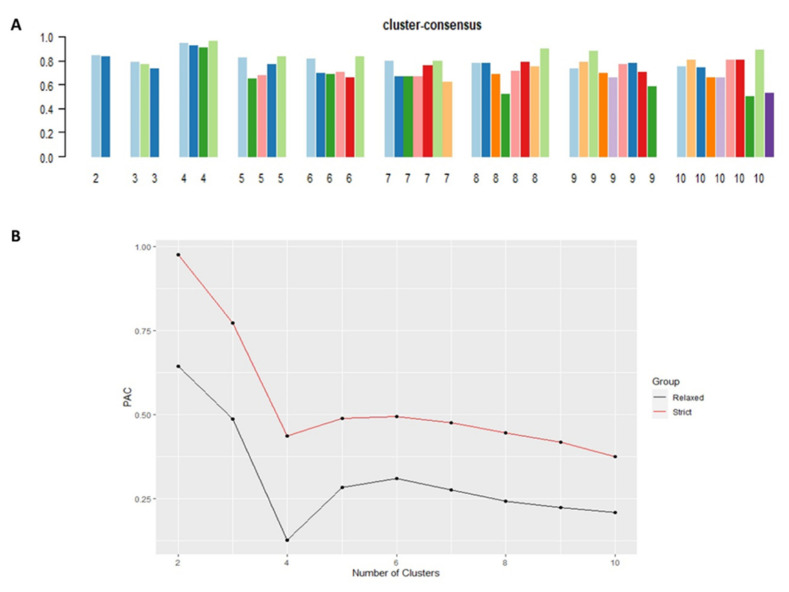
(**A**) The bar plot represents the mean consensus score for different numbers of clusters (K ranges from two to ten); (**B**) Definition for ambiguously clustered pairs utilizing PAC values with the strict criteria (red line) with the predetermined boundary of (0, 1), and the PAC values with the relaxed criteria (black line) with the predetermined boundary of (0.1, 0.9).

**Figure 3 medsci-09-00060-f003:**
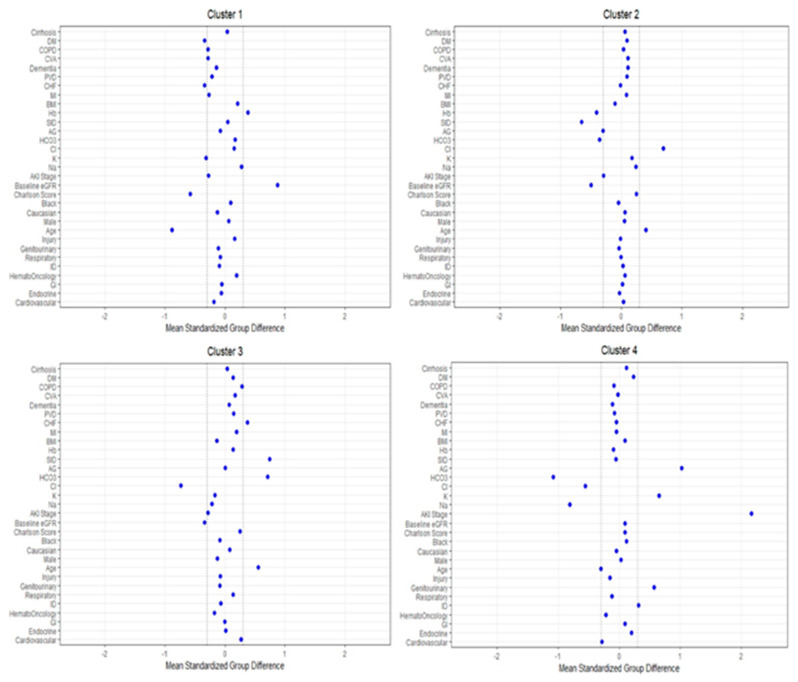
The standardized differences across the four clusters for each baseline parameter. The x-axis is the standardized difference value, and the y axis shows baseline parameters. The dashed vertical lines represent the standardized differences cutoffs of <−0.3 or >0.3. Abbreviations: AKI, acute kidney injury; DM, diabetes mellitus; COPD, chronic obstructive pulmonary disease; CVA, cerebrovascular accident; PVD, peripheral vascular disease; CHF, congestive heart failure; MI, myocardial infarction; BMI, body mass index; Hb, hemoglobin; SID, strong ion difference; AG, anion gap; ESKD, end-stage kidney disease; HCO3, bicarbonate; Cl, chloride; K, potassium; Na, sodium; GFR, glomerular filtration rate; RS, respiratory system; ID, infectious disease; GI, gastrointestinal.

**Figure 4 medsci-09-00060-f004:**
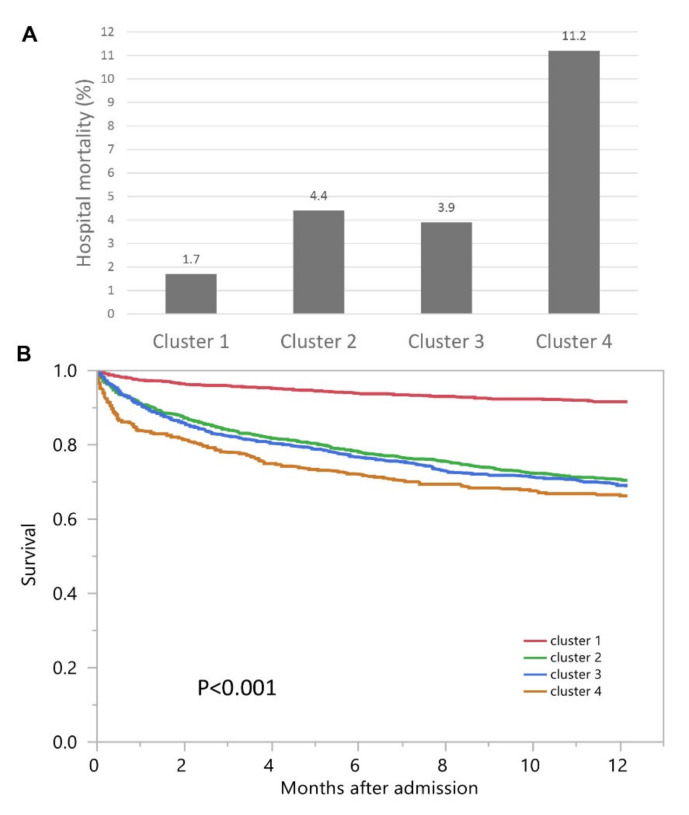
(**A**) Hospital mortality among different clusters with admission AKI; (**B**) One-year mortality among different clusters with admission AKI.

**Table 1 medsci-09-00060-t001:** Baseline clinical characteristics.

Patient Characteristics	Overall	Cluster 1	Cluster 2	Cluster 3	Cluster 4	*p*-Value
	(*n* = 4289)	(*n* = 1201)	(*n* = 1396)	(*n* = 1191)	(*n* = 501)	
Age (years)	67.3 ± 16.2	53.0 ± 14.3	73.8 ± 12.0	76.2 ± 10.9	62.4 ± 15.5	<0.001
Male sex	2566 (60)	754 (63)	869 (62)	637 (53)	306 (61)	<0.001
Race						<0.001
-White	4042 (94)	1096 (91)	1336 (96)	1143 (96)	467 (93)	
-Black	65 (2)	32 (3)	13 (1)	5 (0.4)	15 (3)	
-Others	182 (4)	73 (6)	47 (3)	43 (4)	19 (4)	
BMI (kg/m^2^)	30.4 ± 8.3	32.1 ± 9.8	29.5 ± 7.1	29.3 ± 7.2	31.2 ± 9.5	<0.001
Principal diagnosis						<0.001
-Cardiovascular	820 (19)	141 (12)	287 (21)	352 (30)	40 (8)	
-Endocrine/metabolic	190 (4)	38 (3)	54 (4)	55 (5)	43 (9)	
-Gastrointestinal	437 (10)	103 (9)	149 (11)	120 (10)	65 (13)	
-Genitourinary	499 (12)	98 (8)	145 (10)	106 (9)	150 (30)	
-Hematology/oncology	741 (17)	296 (25)	274 (20)	126 (11)	45 (9)	
-Infectious disease	381 (9)	74 (6)	135 (10)	83 (7)	89 (18)	
-Respiratory	216 (5)	40 (3)	70 (5)	94 (8)	12 (2)	
-Injury/poisoning	488 (11)	198 (16)	152 (11)	105 (9)	33 (7)	
-Other	517 (12)	213 (18)	130 (9)	150 (13)	24 (5)	
Charlson Comorbidity Score	3.0 ± 2.7	1.4 ± 1.7	3.6 ± 2.8	3.7 ± 2.8	3.2 ± 2.9	<0.001
Comorbidities						
-Coronary artery disease	530 (12)	43 (4)	210 (15)	223 (19)	54 (11)	<0.001
-Congestive heart failure	632 (15)	33 (3)	199 (14)	334 (28)	66 (13)	<0.001
-Peripheral vascular disease	272 (6)	12 (1)	121 (9)	117 (10)	22 (4)	<0.001
-Dementia	119 (3)	5 (0.4)	63 (63)	46 (4)	5 (1)	<0.001
-Stroke	518 (12)	34 (3)	218 (16)	209 (18)	57 (11)	<0.001
-COPD	629 (15)	56 (5)	222 (16)	293 (25)	58(12)	<0.001
-Diabetes mellitus	1390 (32)	198 (16)	516 (37)	459 (39)	217 (43)	<0.001
-Cirrhosis	236 (6)	40 (3)	90 (6)	47 (4)	59 (12)	<0.001
Laboratory test						
-eGFR (mL/min/1.73 m^2^)	68 ± 27	92 ± 23	55 ± 20	59 ± 21	71 ± 29	<0.001
-Sodium (mEq/L)	137 ± 5	138 ± 4	138 ± 4	136 ± 5	133 ± 6	<0.001
-Potassium (mEq/L)	4.5 ± 0.8	4.3 ± 0.6	4.7 ± 0.8	4.4 ± 0.7	5.0 ± 1.0	<0.001
-Chloride (mEq/L)	102 ± 6	103 ± 4	106 ± 4	98 ± 5	99 ± 7	<0.001
-Bicarbonate (mEq/L)	24 ± 5	25 ± 3	22 ± 4	27 ± 4	19 ± 5	<0.001
-Anion gap	11 ± 4	10 ± 3	9 ± 3	11 ± 4	15 ± 6	<0.001
-Strong ion difference	39.2 ± 4.3	39.4 ± 3.2	36.3 ± 3.4	42.4 ± 3.5	38.9 ± 5.2	<0.001
-Hemoglobin (g/dL)	11.6 ± 2.3	12.5 ± 2.2	10.7 ± 2.1	11.9 ± 2.0	11.4 ± 2.6	<0.001
Acute kidney injury stage						<0.001
-Stage 1	3517 (82)	1092 (91)	1289 (92)	1092 (92)	44 (9)	
-Stage 2	408 (10)	102 (8)	93 (7)	86 (7)	127 (25)	
-Stage 3	364 (8)	7 (1)	14 (1)	13 (1)	330 (66)	

**Table 2 medsci-09-00060-t002:** Mortality outcomes according to clusters.

	Hospital Mortality	OR (95% CI)	1-Year Mortality	HR (95% CI)
Cluster 1	1.7%	1 (ref)	8.4%	1 (ref)
Cluster 2	4.4%	2.74 (1.65–4.57)	29.7%	3.97 (3.14–5.03)
Cluster 3	3.9%	2.37 (1.39–4.04)	31.2%	4.22 (3.33–5.35)
Cluster 4	11.2%	7.43 (4.41–12.53)	33.7%	4.98 (3.82–6.48)

## Data Availability

Data is available upon reasonable request to the corresponding author.

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
