# Peer review of "Clinically Distinct Subtypes of Acute Kidney Injury on Hospital Admission Identified by Machine Learning Consensus Clustering"

_medsci, 2021, doi:10.3390/medsci9040060_

Round 1
Reviewer 1 Report
The purpose of the study was, based on laboratory data from patients with acute kidney injury at hospital to evaluate the short- and long-term prognosis of mortality. Patients with hyponatremia were identified through an unsupervised learning machine approach and divided into related cluster. Four main groups have been identified. Cluster 1: patients younger with highest eGFR and hemoglobin on admission. Cluster 2: patients had lower serum bicarbonate, strong ion difference, and hemoglobin, but higher serum chloride, whereas cluster 3 patients had lower serum chloride but higher serum bicarbonate and strong ion difference. Cluster 4 patients were younger and more likely to be admitted for genitourinary disease and infectious disease but less likely to be admitted for cardiovascular disease. Cluster 4 patients also had more severe acute kidney injury, lower serum sodium, serum chloride, and serum bicarbonate, but higher serum potassium and anion gap. Cluster 2, 3, and 4 patients had significantly higher hospital and one-year mortality than cluster1 patients (p<0.001). As former paper analyzing the same population with hyponatremia, this one is also interesting. Some observations must be done. 1. The abbreviation of NAGMA (non-amnio-gap metabolic acidosis). 2. In the summary, cluster 1 should be more characterized by including two more findings as highest eGFR and hemoglobin. Maybe it would explain the lowest mortality? 3. The higher mortality of cluster 4 is attributed to the gravity of the infections or for the higher degree of hyponatremia? The paper should be accepted with minor reviews.
Author Response
Response to Reviewer#1
Comment
The purpose of the study was, based on laboratory data from patients with acute kidney injury at hospital to evaluate the short- and long-term prognosis of mortality. Patients with hyponatremia were identified through an unsupervised learning machine approach and divided into related cluster. Four main groups have been identified. Cluster 1: patients younger with highest eGFR and hemoglobin on admission. Cluster 2: patients had lower serum bicarbonate, strong ion difference, and hemoglobin, but higher serum chloride, whereas cluster 3 patients had lower serum chloride but higher serum bicarbonate and strong ion difference. Cluster 4 patients were younger and more likely to be admitted for genitourinary disease and infectious disease but less likely to be admitted for cardiovascular disease. Cluster 4 patients also had more severe acute kidney injury, lower serum sodium, serum chloride, and serum bicarbonate, but higher serum potassium and anion gap. Cluster 2, 3, and 4 patients had significantly higher hospital and one-year mortality than cluster1 patients (p<0.001). As former paper analyzing the same population with hyponatremia, this one is also interesting. Some observations must be done.
Response: We thank you for reviewing our manuscript and for your critical evaluation.
Comment #1
The abbreviation of NAGMA (non-amnio-gap metabolic acidosis)
Response: We agree with the reviewer. This change has been made as suggested.
Comment #2
In the summary, cluster 1 should be more characterized by including two more findings as highest eGFR and hemoglobin. Maybe it would explain the lowest mortality?
Response: We agree with the reviwer. We added this important point in the discussion on the lowest mortality among patients in cluster 1 as suggested.
“Patients in cluster #1 had the lowest mean age and comorbidity burden while simultaneously having the highest eGFR and hemoglobin, which could explain the lowest mortality in this cluster”
Comment #3
The higher mortality of cluster 4 is attributed to the gravity of the infections or for the higher degree of hyponatremia? The paper should be accepted with minor reviews
Response: We agree with the reviewer. We added this important point in the discussion as suggested.
“Although clusters #2 and #3 were older and had higher comorbidities than those in cluster #4, patients in clusters #4 carried higher in-hospital and one-year mortality risks. This is likely represents due to the significant impacts of AKI severity, infectious disease, and hyponatremia on poor outcomes [32,33]”
Thank you for your time and consideration. We greatly appreciated the reviewer’s and editor’s time and comments to improve our manuscript. The manuscript has been improved considerably by the suggested revisions.
Reviewer 2 Report
Dear Authors,
your article seems quite interesting and enhances the possibles benefits from the application of machine learning in medicine
Few points needs to be furtherly elucidated:
1) materials and methods: How was data on subsequent hospitalization or follow-up collected? How mortality was evaluated? It need to be specified, just to have better information on patients' lost at follow-up, who might be both alive or dead
2) materials and methods: how long was the eventual follow-up period?
3) results: report lost at follow-up rate and compare them between groups
4) discussion: better describe differences and possible pathological pathways which define the different clusters
Author Response
Response to Reviewer#2
Comment
Your article seems quite interesting and enhances the possible benefits from the application of machine learning in medicine. Few points needs to be furtherly elucidated
Response: We thank you for reviewing our manuscript and for your critical evaluation.
Comment #1
Materials and methods: How was data on subsequent hospitalization or follow-up collected? How mortality was evaluated? It need to be specified, just to have better information on patients' lost at follow-up, who might be both alive or dead
Response: The reviewer raises important point. We agree and the following statement have been added to the method section.
“The outcomes were hospital mortality and 1-year mortality. We initiated follow-up from hospital admission until death or 1-year after hospital admission and censored their follow-up at the date of their last inpatient/outpatient follow-up visit. We determined patient death and their death date using our hospital’s registry and Social Security Index.”
Comment #2
materials and methods: how long was the eventual follow-up period?
Response: The median follow-up time was 1.1 (IQR 0.3-2.1) years. We added this information in our manuscript as suggested.
Comment #3
results: report lost at follow-up rate and compare them between groups
Response: We appreciate the reviewer’s important comment. We have complete follow-up for hospital mortality, allowing us to use logistic regression to comparing mortality outcomes among clusters. Meanwhile, we conducted time-to-event analysis using Cox proportional hazard regression for 1-year mortality to account for different follow-up time of each patient. In this analysis, if death outcome is not known before the end of follow-up, patient is censored at their last follow-up visit. Also, we used Social Security Death Index, in addition to our hospital’s registry, to capture death elsewhere.
Comment #4
discussion: better describe differences and possible pathological pathways which define the different clusters
Response: We appreciate the reviewer’s input. We agree and thus additionally revised our discussion as suggested.
Thank you for your time and consideration. We greatly appreciated the reviewer’s and editor’s time and comments to improve our manuscript. The manuscript has been improved considerably by the suggested revisions.
Reviewer 3 Report
Thanks for the invitation to review this paper. I have carefully reviewed the manuscript entitled " Clinically Distinct Subtypes of Acute Kidney Injury on Hospital Admission identified by Machine Learning Consensus Clustering" (medsci-1323696) by Charat Thongprayoon et al. This single-center, retrospective study enrolled 4,289 adult hospitalized patients with CA-AKI admitted to Mayo Clinic Hospital in Rochester, Minnesota, from January 2011 to December 2013. Using the unsupervised consensus clustering analysis, the authors categorized all patients into four clusters with different clinical characteristics and subsequently demonstrated that the four clusters had different outcomes (in-hospital mortality and 1-year mortality). Generally speaking, this is a well-written work. The English writing, including expression, fluency, and readability, of this manuscript, is good. The study design and results are technically sound. Nevertheless, I have some concerns regarding this paper. # My primary concern is “what is the main take-home message of the paper for the readers?” The authors applied machine learning to categorize four clusters with varied clinical characteristics from a single-center cohort. If we apply the same machine learning in a different cohort, the results (including the cluster numbers and the characteristics of the four clusters) might be different from the current study’s results. To a new beginner like me, what this study demonstrated is that "machine learning could cluster patients with clinically distinct characteristics." I do not whether I got the actual contents of the study, but I suggest the authors make more address on the strength of the study. # Abbreviation is not applied throughout the manuscript (ex: AKI) # Why do the authors choose to apply the cohort 2011-2013 rather than another cohort that was more updated?Author Response
Response to Reviewer#3
Comment
Thanks for the invitation to review this paper. I have carefully reviewed the manuscript entitled " Clinically Distinct Subtypes of Acute Kidney Injury on Hospital Admission identified by Machine Learning Consensus Clustering" (medsci-1323696) by Charat Thongprayoon et al. This single-center, retrospective study enrolled 4,289 adult hospitalized patients with CA-AKI admitted to Mayo Clinic Hospital in Rochester, Minnesota, from January 2011 to December 2013. Using the unsupervised consensus clustering analysis, the authors categorized all patients into four clusters with different clinical characteristics and subsequently demonstrated that the four clusters had different outcomes (in-hospital mortality and 1-year mortality). Generally speaking, this is a well-written work. The English writing, including expression, fluency, and readability, of this manuscript, is good. The study design and results are technically sound. Nevertheless, I have some concerns regarding this paper.
Response: We thank you for reviewing our manuscript and for your critical evaluation.
Comment #1
My primary concern is “what is the main take-home message of the paper for the readers?” The authors applied machine learning to categorize four clusters with varied clinical characteristics from a single-center cohort. If we apply the same machine learning in a different cohort, the results (including the cluster numbers and the characteristics of the four clusters) might be different from the current study’s results. To a new beginner like me, what this study demonstrated is that "machine learning could cluster patients with clinically distinct characteristics." I do not whether I got the actual contents of the study, but I suggest the authors make more address on the strength of the study
Response: We appreciate the reviewer’s important comment. We agree and thus included more strengths and summary in the conclusion of our revised manuscript as suggested.
“Highest mortality risk of AKI on admission was observed among patients with higher AKI severity, hyponatremia, metabolic acidosis, and principal diagnosis of infectious disease. These findings may potentially help classify hospitalized patients with AKI on admission which are associated with different mortality risks, and translate towards an improved personalized medicine approach.”
Comment #2
Abbreviation is not applied throughout the manuscript (ex: AKI)
Response: We appreciate the reviewer’s comment. We agree and this change has been made as suggested.
Comment #3
Why do the authors choose to apply the cohort 2011-2013 rather than another cohort that was more updated?
Response: We agreed with your suggestion. However, our dataset in more recent years is still incomplete, and therefore, contains less comprehensive information. The following statements have been added to the limitation.
“In addition, as this study was conducted using the data of hospitalized patients from 2011 to 2013, future studies using a more up-to-date dataset is suggested to confirm our finding”
Thank you for your time and consideration. We greatly appreciated the reviewer’s and editor’s time and comments to improve our manuscript. The manuscript has been improved considerably by the suggested revisions.